# Mental Health and Body Image and the Reduction of Excess Body Weight in Woman (Polish Sample)

**DOI:** 10.3390/nu16060811

**Published:** 2024-03-13

**Authors:** Hanna Liberska, Klaudia Boniecka

**Affiliations:** 1Faculty of Psychology, Kazimierz Wielki University, ul. Chodkiewicz 30, 85-064 Bydgoszcz, Poland; 2Independent Researcher, 85-064 Bydgoszcz, Poland; klaudia.boniecka@gmail.com

**Keywords:** diet, physical activity, weight reduction, body image, mental health, woman

## Abstract

In recent years, excessive body weight has become one of the most serious psychological, biological and social problems. According to forecasts by the World Health Organization, obesity and overweight will continue to increase among both adults and children in the coming years. Poland ranks sixth in Europe in this respect. In 2021, almost 58% of Poles had above-average body weight (overweight or obesity). In Poland, 29% of women are overweight, and 21% of women are obese. Body dissatisfaction, depression, and anxiety disorder are indicated as consequences of high BMI in women. Reducing excess body weight improves psychosocial functioning and changes body assessment. The most lasting effects of weight reduction are achieved by a properly selected diet combined with increased physical activity. This results in a change in lifestyle, changes in the levels of metabolic indicators, and changes in one’s body image and mental health. Objective: Our objective was to assess changes in body image and mental health among Polish women and their dependence on the weight reduction method used (type of diet and physical activity). Comparative studies involving 122 women were conducted. These women were selected from 1000 volunteers based on BMI criteria. The effects of diet therapy were compared with the effects of diet therapy combined with physical activity. Research tools: The GHQ-12 scale was used to measure mental health, and the Body Esteem Scale was used to measure body image. The study lasted four years. The results showed changes in body image, general mental health index, and BMI in women who dieted and in those who dieted and exercised. In women using a diet combined with physical activity, greater positive changes in mental condition and stronger positive changes in body image, namely in the assessment of satisfaction with sexual attractiveness, physical condition, and body weight control, were observed compared to women using diet therapy alone.

## 1. Introduction

In recent years, excess body weight has been one of the most serious psychological, biological, and social problems affecting many populations [1,2,3]. According to the forecasts of the World Health Organization, in the coming years, there will be a further increase in obesity and overweight among both adults and children [4]. Data from the latest update of the Global Food Security Index confirm this trend. This report points to the increased weight gain among European citizens. Recently, the prevalence of obesity in adults in Western European countries has exceeded 25%, thus approaching pandemic proportions. According to research, Poland ranks sixth in Europe in terms of obesity rate (defined as the percentage of the population over 20 years old with a BMI > 30), with rate that is currently at 23.2 percent [5]. The risk of obesity has increased in the Polish population in the period from 2004 to 2022. In 2022, more than 9 million Poles struggled with obesity [6]. The risk of obesity increases significantly in Polish women over 45 years of age [7]. One of the consequences of a high BMI in women is body dissatisfaction [8,9], as well as women’s fears of negative evaluation. However, some results have not confirmed the negative correlation of body mass index with fear of negative evaluation [10]. 

This is a multifaceted problem, because excess body weight is associated with depression, anxiety disorders, and problems related to emotional functioning [11,12], which may contribute to the further deterioration of one’s mental state.

Both obesity and depression significantly reduce the quality of life of ill people and contribute to increased mortality [13]. However, the links between diet, mental health, and stress are not fully understood [14]. Some studies suggest that symptoms of depression in obese women are closely related to body dissatisfaction [15]. The discrepancy between the ideal body image and the real body image may lead to dissatisfaction with one’s body and, consequently, may be a risk factor for mental deterioration in people with excess body weight [16,17].

The perceived health condition and satisfaction with physical appearance are related to the image of oneself, a component of which is the image of one’s own body [18]. Self-image, including the image of one’s own body, is assumed to be a regulator of psycho-social functioning [19]. Numerous studies have shown that women are more likely than men to be dissatisfied with their own bodies [20]. Young women are more susceptible to media messages promoting a slim figure, and thus, in their mind, there is more often a discrepancy between the assessment of the body they have (body in self-perception) and the body they would like to have [21].

Even a slight decrease in body weight brings an improvement in metabolic indicators such as lipid levels, glucose levels, and blood pressure [22]. A 10% reduction in body weight has been shown to contribute to a significant and lasting improvement in health [23]. The results of the research conducted so far in this area provide grounds for assuming that the reduction of excess body weight also improves psycho-social functioning and changes one’s body assessment [1,24,25,26,27]. 

Many researchers emphasize the fact that the most effective and durable method of reducing excess body weight is the use of properly selected diet therapy, along with increased physical activity [2]. Knowledge about new, effective dietary, pharmacological, and preventive interventions is increasing [28]. Understanding the relationship between body weight, a rational diet, and physical activity adapted to somatic and mental health provides the basis for psychocorrective effects based on a healthy lifestyle, including the introduction of psychobiotics and antioxidants into food [29]. This may result in improvements in one’s emotional state and body image. The use of weight reduction methods with scientifically unproven effectiveness is dangerous to the health of people with excess body weight. Sometimes, the use of these methods is the result of one succumbing to imprecise media messages or pressure from one’s immediate environment, including a life partner, and these methods may contribute to the deterioration of one’s health, as well as negative changes in body image and satisfaction.

To the best of our knowledge, there are no longitudinal studies verifying differences in mental health and body assessment before and after effective weight loss, including the method of weight loss, in the population of Polish women. Mental health is defined by the WHO as a state of mental well-being that enables people to cope with the stresses of life, realize their abilities, learn well and work well, and contribute to their community [30,31]. Therefore, longitudinal research was undertaken to identify changes in the body image and mental health of Polish women depending on the weight loss method.

The following research questions were formulated: (1) Is there is a difference in mental health and body assessment before and after effective weight reduction? (2) Do the results of women who choose to reduce excess body weight based on diet and physical activity differ from those of women using short-term dietary interventions in terms of mental health assessment and body assessment? (3) Is there is a relationship between mental health assessment and body assessment? 

## 2. Materials and Methods

### 2.1. Participants 

A total of 122 women aged from 20 to 37 (mean age: 29.16) participated in the study. The selection of the respondents for the sample was deliberate. The following criteria were used in the selection of people for the sample: the presence of excessive body weight (not resulting from hormonal or genetic disorders, CNS damage, or taking medications), reduction of excessive body weight obtained by the selected method of weight reduction, gender, age. In order to identify women with excess body weight, it was decided to use the Queteleta II index (body mass index) (despite the awareness of the limitations associated with its use) because of its ease of use and its use in many studies [32,33]. Overweight and obese women were treated as one group, i.e., as people with excess body weight, because of the small group of respondents. Due to the great difficulty in finding an appropriate number of women meeting the adopted criteria, it was decided to use a non-random sampling method, namely the snow ball sampling method [34]. The examined women recommended to the researcher other persons who met the criteria and could be examined after giving their consent to take part in the study. 

### 2.2. Measures

The following tools were used in the study: the Body Esteem Scale developed by S.L. Franzoi and S.A. Shields [35] (specifically the Polish adaptation of Lipowska and Lipowski) [36] and a shortened version of the General Health Questionnaire GHQ-12 by D. Goldberg [37] (again, we used the Polish adaptation of Z. Makowska and D. Merecz [38]). A survey prepared for the needs of the study was also used. This survey provided data on the treatment of excess body weight and criteria related to the women tested, such as the chosen method of weight loss, age, education, body weight, and height. 

General Health Questionnaire GHQ-12 [37], in Polish adaptation: Z. Makowska and D. Merecz [38]. 

Mental health was assessed on the basis of the results obtained in the shortened version of the General Health Questionnaire. GHQ-12 is used to assess symptoms of mental health and allows for the identification of people whose mental state has been subject to a temporary or long-term breakdown as a result of difficulties or problems they have experienced. The questionnaire can be used to assess changes in a person’s mental state over time [38]. The sum of the points obtained for answering all the questions indicates your overall mental health score. The higher the score, the worse your mental health is. In the Polish adaptation, high internal compliance coefficients and a satisfactory absolute stability were obtained [38]. Reliability indices (Cronbach’s alpha) for the overall score can range from 0.79 to 0.89 depending on the group of respondents. The correlation coefficient between duplicate tests is 0.68. 

Body Esteem Scale S.L. Franzoi and S.A. Shields [35], Polish adaptation: M. Lipowska and M. Lipowski [36]). 

This scale is used to study body image and is often used in studies of obese and overweight people [36,39]. This scale includes 35 items that form three sub-scales—different for men and women. The sub-scales for women measure sexual attractiveness, weight control, and physical condition. The sub-scales used for men include physical attractiveness, body strength, and physical condition. The higher the score on a given scale, the greater the body satisfaction in that area. The reliability of the original version of the method is satisfactory. The Body Esteem Scale’s reliability index (Cronbach’s alpha) in the part for women—depending on the sub-scale—is valued to be in the range of <0.78; 0.87> [35]. The results of the analysis of the reliability of the Polish version for women are as follows: the Cronbach’s alpha scale of sexual attractiveness = 0.80, the Cronbach’s alpha scale of physical condition = 0.89, the Cronbach’s alpha scale of physical condition = 0.82. For men, the scale of physical attractiveness Cronbach’s alpha = 0.85, the body strength scale Cronbach’s alpha = 0.85, and the physical condition scale Cronbach’s alpha = 0.88 [36]. 

### 2.3. Procedure 

The studies were longitudinal studies and took place in two stages. The first stage lasted from January 2015 to September 2016. The procedure of selecting a group of research subjects began with an examination of about 1000 women who met the adopted criteria and wanted to start slimming therapies. Based on the input data (height and weight), body mass index-1 (BMI-1) was obtained. The diagnosis of obesity and overweight is based on the analysis of body weight and the location of fat tissue. To gain accurate measurements of fat tissue in the organs and the retroperitoneal space, imaging methods are used (computed tomography (CT) and magnetic nuclear resonance (MRI)), along with bioelectrical impedance analysis (BIA) or dual-energy X-ray absorptiometry (DEXA). However, these tests are time-consuming and expensive. Therefore, anthropometric methods are used in population studies which use, for example, height and body weight or hip and waist circumference, allowing for the calculation of anthropometric indicators. The most frequently used indicators in adults include the BMI, WHR, WHtR, and WC [40]. The most popular one is the BMI (body mass index), also called the Quetelet index. One of the newest anthropometric indicators is the body adiposity index (BAI). It is a response to the imperfections of the BMI because it allows for a better assessment of the risk of health disorders due to the determination of body fat content. The BAI was proposed in 2011 after research was conducted to find out the characteristics that had the strongest association with obesity diagnosed by X-ray densitometry among African American and Mexican populations. The BAI is calculated as the ratio of hip circumference to body height according to the following formula: BAI = (hip circumference/(height)^1.5^) − 18 [41]. A limitation of BAI is its decreased measurement accuracy in cases of morbid obesity. Many studies have confirmed that the relationship between the BAI and obesity is stronger than that between the BMI and obesity [42,43]. However, some studies in Caucasian populations contradict these reports. Geliebter, Atalayer, and Flancbaum compared results obtained using the BMI and BAI to results derived from examining the body fat content of morbidly obese people using the DEXA method, previously described in [44]. It turned out that the BMI correlated with the DEXA results, while in the case of the BAI, there was no correlation. Also, Vinknes et al. reported a higher correlation of BMI with the results obtained using the DEXA method after dividing the subjects by gender [45]. As can be seen from the above, both anthropometric indices, BMI and BAI, have limitations, but due to their ease and speed of measurement, they are important tools in epidemiological and clinical research and in practice. 

For the above reasons, including due to its ease and speed of measurement, the BMI was chosen.

In the first stage, the subjects were classified into groups based on their chosen method of weight reduction. Based on the choices of the respondents, four methods of reducing excess body weight were distinguished: diet therapy with increased physical activity, diet therapy with increased physical activity under the supervision of a specialist, using a low-energy diet, and using an elimination diet [28,29,32]. It should be noted that the first two methods are aimed at long-term lifestyle changes, while the last two methods are interventions aimed solely at reducing excess body weight. On this basis, the women were classified into groups using diet therapy combined with increased physical activity (some of whom were under the care of a qualified trainer and some did not benefit from such support) or into groups using only a low-energy diet or elimination diet therapy (Figure 1). The first stage of the examination was carried out from 1 to 7 days before the start of the slimming treatment.

The research implemented in the second stage was aimed at obtaining answers to the research questions. The second stage was carried out after the examined women achieved the correct body weight reported in the first stage (after effective weight reduction from 7 to 14 days). It lasted from October 2016 to October 2019. Finally, 122 women who managed to reduce their excess body weight to the established norms were examined. The indicator of success in weight reduction was the calculated BMI-2. The study involved people who did not report eating disorders and did not participate in psychotherapy before starting the weight loss process, neither in the first nor in the second stage of the study. The examined women received a set of questionnaires with instructions and a request to complete and return them within the specified time. The respondents were assured their responses would be anonymous. 

It should be emphasized that the duration of therapy varied individually. Losing weight took different amounts of time for individual women. Some women achieved the desired BMI after only 4–5 months of the reduction process, while for other women, the reduction process lasted 3 years, and some women did not achieve the goal of achieving the correct BMI during the research period. Not all women who expressed their willingness to participate in the study in the first stage of the study (as mentioned above) persisted in their intention to reduce body weight. The declared reasons for resignation were various: medical indications, pregnancy, loss of motivation, change in place of residence, resignation without giving a reason, etc.

This study was approved by the Research Ethics Committee of the Kazimierz Wielki University.

### 2.4. Data Analysis 

The collected results were statistically analyzed using STATISTICA version 6.0. The parameter values are presented herein in the form of the arithmetic mean (M) and standard deviation (SD). To compare the mean values, Student’s *t*-test was used for both the dependent and independent groups, assuming statistically significant differences at the level of *p* ≤ 0.05. Pearson’s r correlation coefficient was used to assess the correlation of the results in both groups. The Mann–Whitney U test and stepwise regression analysis were also used. An ANOVA was not used in our analysis due to the requirement for random sampling [46].

## 3. Results

### 3.1. Differences between the Means in Terms of the Studied Variables before and after the Start of the Weight Loss Therapies 

In order to check the significance of the differences, Student’s *t*-test was used. The results of the analysis are presented in Table 1. 

It was found that the BMI of the examined women differed significantly in the first and second measurement (t = 23.77; *p* < 0.001). In the first measurement, i.e., before participating in the weight loss therapies, the examined women had a higher BMI (M = 29.94; SD = 3.72) than after the weight loss therapies (M = 22.35; SD = 1.90). The means in the first measurement indicated overweight, while the means in the second measurement indicated normal weight.

Regarding the measurements derived from using the General Health Questionnaire (GHQ-12), significant differences were found between the first and the second measurement (t = −8.36; *p* < 0.001). It turned out that the examined women obtained higher results in the first measurement (M = 31.64; SD = 5.31) compared to the second measurement (M = 24.89; SD = 6.31). This indicates that in the second measurement, the surveyed women reported significantly fewer problems in terms of mental health than in the first measurement.

Regarding body assessment, statistically significant differences in the examined women were found between the results before the start of weight loss therapy and the results obtained after weight loss in the assessment of sexual attractiveness (t = −11.84; *p* < 0.001), weight control (t = −13.44; *p* < 0.001), and physical condition (t = −13.44; *p* < 0.001). The examined women considered their sexual attractiveness to be significantly higher after losing excessive body weight (M = 51.07; SD = 7.11) than it was before they started weight loss therapy (M = 37.85; SD = 9.71). The examined women considered their weight control to be better after losing weight (M = 38.74; SD = 6.78) than it was before they started the weight loss therapies (M = 24.56; SD = 8.96). In the case of the first measurement, the respondents assessed their physical condition as lower (M = 24.82; SD = 7.40) than in the second measurement (M = 36.63; SD = 4.98). 

### 3.2. Differences between the Studied Variables in Two Separate Study Groups of Women before and after Weight Loss Therapies

In our statistical analysis Student’s *t*-test and the Mann–Whitney U test were used. Statistical analyzes were aimed at providing a basis for obtaining an answer to the question about the existence of differences in the assessment of the body and mental health in the surveyed women before starting weight loss therapies and after completing the therapy, depending on the chosen weight loss method. The first group included 60 women, while the second group consisted of 62 women. The first group consisted of women who achieved a reduction in excess body weight through long-term dietary changes and the introduction of physical activity, while the second group of respondents were women who used various types of short-term diets to minimize excess body weight. The results of the analysis are presented in Table 2 and Table 3. 

The results of the statistical analysis showed no significant differences for any of the variables studied prior to the initiation of weight reduction. On the other hand, statistically significant differences occurred in the second measurement, i.e., after effective weight reduction. These differences occurred in the assessment of mental state (z = −3.77; *p* < 0.001). The women who reduced their excess body weight through long-term dietary changes and the introduction of physical activity (group 1) reported significantly fewer mental health problems than the women who used short-term psychological interventions for weight reduction (group 2). Significant differences were also noted in the level of sexual attractiveness assessment (t = 6.15; *p* < 0.001)—the women using diet therapy with physical activity (group 1) achieved a higher level of satisfaction with their sexual attractiveness (M = 54.60) than the women (group 2) using short-term dietary interventions (M = 47.66); in the weight control assessment (z = 5.80; *p* < 0.001), the women from group 1 (M = 42.02) presented a higher level of satisfaction with weight control than the women from group 2 (M = 38.87), and in terms of satisfaction with physical condition (z = 5.10; *p* < 0.001), the women from group 1 obtained higher results on this scale (M = 38.86) than the women from group 2 (M = 34.47). 

The differences between the variables are presented graphically in Figure 2.

### 3.3. Relationships between Body Assessment and the Level of Mental Health in the Group of Surveyed Women

The next part of the statistical analysis focused on examining the relationship between body assessment and mental health levels among the women before weight loss (measurement 1) and after the reduction of excess body weight (measurement 2). For this purpose, Pearson’s r correlations between the above-mentioned variables were calculated (Table 4). 

A significant relationship between the assessment of the body in its individual aspects and the level of mental state was found only in the second measurement. In order to clarify which categories of body assessment significantly explain the variance of mental state, step regression analyses (only for the second measurement) were performed. In each regression analysis, all aspects of body satisfaction were entered as independent variables, while general mental health was entered as the dependent variable (Table 5).

For the mental health variable, two body-rating sub-scales turned out to be sufficient for explaining the variance. In the second measurement, weight control (beta = −0.227; *p* < 0.05) and physical condition (beta = −0.344; *p* < 0.001) proved to be statistically significant predictors of mental health (F = 20.49; *p* < 0.001), which explains the 24.4% variation in the mental health results.

## 4. Discussion 

The results of our quantitative analysis show that before the reduction of excessive body weight, the examined women were characterized by higher rates of abnormal body mass (BMI), a higher level of problems with their mental states, and an inferior body assessment. The results we obtained are consistent with the assumptions and confirmed results presented in the literature on this subject [19,23,24]. They correlate with both bio-psychological theories pointing to the role of dietary changes in human functioning [47,48,49] and views on body image and its impact on psycho-social functioning [17,19,26]. The improvement in the general mental health index obtained in this study also corresponds to the results of other studies, according to which, after obtaining a normal body weight, there is a significant improvement in the mental state [31,50,51,52,53]. In line with the assumed hypothesis, a significant increase in body assessment was noted in all the tested sub-scales. Increased satisfaction with one’s own body after weight loss is one of the determinants of a correct adaptation to one’s new body weight [50].

It has been shown that there are significant differences in body assessment and mental health levels between women who have reduced their excess body weight through diet and exercise and those who have applied short-term dietary interventions. Women from the first group, i.e., those using diet and physical activity, reported significantly fewer problems with mental health than women from the second group and had a higher self-assessed rating of sexual attractiveness, physical condition, and weight control. Current research shows that the most effective method of weight reduction is diet therapy with increased physical activity [51]. Women who use the correct methods of weight reduction function much better [39,52]. Thus, it seems justified to increase the awareness of society in this regard, especially among young girls, who very often use behaviors that are unfavorable to their health in order to achieve their dream body weight [50]. The current focus on human physicality implies the need to take care of the body. However, as shown by numerous studies, this care is often limited to achieving and maintaining a certain body weight and taking care of one’s external appearance [19,21,53]. On the other hand, some studies show that caring for appearance is often not associated with concern for health [12,32,50]. The consumption of highly processed food and fast food and the use of alternating fasting and diets excluding certain groups of food products may result in the achievement of a low body weight but is not good for the somatic and psychological health of the individual. 

It was found that the higher the level of sexual attractiveness, weight control, and physical condition of the women who had lost weight, the fewer the number of mental health problems experienced. The relationship between body assessment and mental state is relatively well understood [29,53]. Body image interventions can improve both mental health and overall weight management efforts [51]. However, not all studies confirm that body dissatisfaction is related to mental health problems. The result that is noteworthy is the lack of relationships in the first measurement, i.e., among the women before weight loss. This allows for the conclusion that not all women with excess body weight who are dissatisfied with their body show problems with their mental state. This corresponds with the results of a study by Jorm et al. [54], who found that body dissatisfaction and mental problems do not occur in all people with excess body weight. Ross and Bird [52] examined mental health problems in the context of slimming diets rather than excess body weight itself. However, the results of some studies on the psychological consequences of bariatric surgery indicate that patients may have difficulties in processing the mental image of their body and low self-esteem, as well as serious symptoms of depression symptoms [55]. On this basis, it can be assumed that the rate of excessive body weight reduction may be related to the difficulties in adapting to the changes taking place in the body and one’s attitude towards it, as well as one’s expectations and fears regarding a radical change in social evaluation after weight loss. Working on your mental body image is a gradual process, and rapid, radical change can be traumatic.

The research presented in this article provides grounds for supposing that a slow weight loss process facilitates the updating of one’s body image representation and contributes to the improvement of self-esteem and mental health.

## 5. Limitation

Of course, the presented research has some limitations. One of the limitations of this study lies in the method used to select people for the sample—the snowball method. The risk associated with this selection method results from the fact that the sample may include people with similar characteristics. In this study, the aim was to include women with excessive body weight in the sample. The snowball method is useful when the research problem concerns people who are stigmatized or excluded in some way—in this case because of their body weight [56]. The data were obtained with the use of questionnaires, so the data consisted of self-reports from the surveyed women, and conclusions about the relationships were drawn based on these data. Despite this study’s numerous advantages, such as the duration of the study and the low cost of conducting the research, the questionnaire methods used were also burdened with disadvantages. First, the respondent could only choose from proposed, ready-made answers [57]. Another problem, especially when it comes to examining the process of treating excess body weight, is drawing conclusions based on one’s declared and not actual health and psycho-social situation. Thus, in future research, it would be valuable to use different measurement methods to obtain more reliable results. The introduction of biochemical tests to monitor the state of the study subjects seems to be an extremely promising notion. 

Due to the extensive effects of the COVID-19 pandemic, which are still being recognized, it is justified to continue longitudinal research on the problem of the relationship between changes in women’s mental health and body image and therapies for reducing excessive body weight.

## 6. Conclusions

The results of this study allow for a broader consideration of the importance of diet and physical activity in the process of treating excess body weight [48]. It is so important to emphasize these behaviors to ensure proper physical and psycho-social functioning instead of only focusing on a specific body weight. The culturally conditioned linking of low body weight with proper functioning is particularly evident nowadays. Many people believe that a good appearance is equivalent to taking care of yourself. However, as numerous authors show, the health behavior of today’s societies is alarmingly incorrect [27,49,53].

## Figures and Tables

**Figure 1 nutrients-16-00811-f001:**
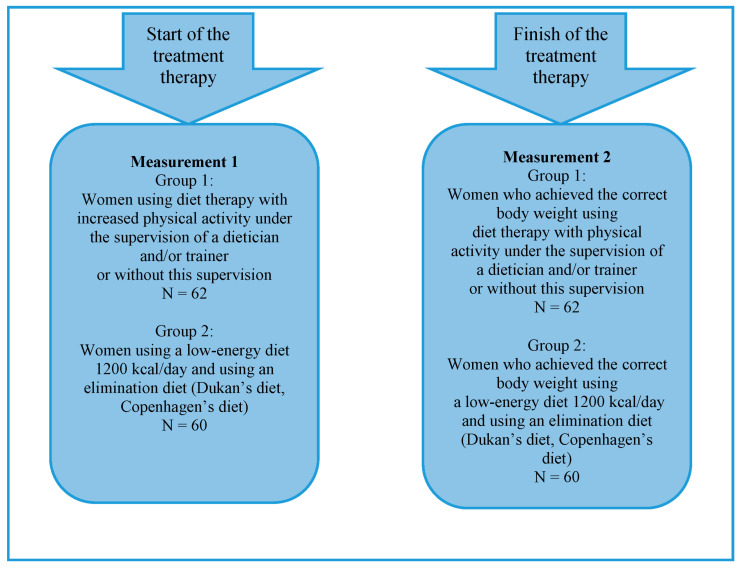
The method of conducting the research.

**Figure 2 nutrients-16-00811-f002:**
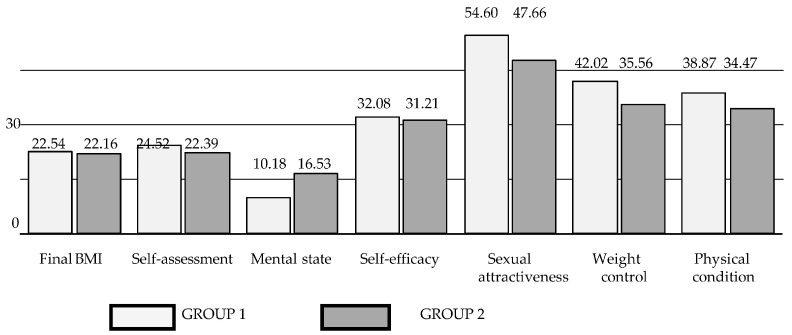
Mean values of measured variables in two groups in the second measurement.

**Table 1 nutrients-16-00811-t001:** The significance of the differences between the means for the analyzed variables in the first and second measurement.

Analyzed Variables	Measurement 1 N = 122		Measurement 2 N = 122		Difference	t	*p*
	M	SD	M	SD			
BMI	29.94	3.72	22.35	1.90	7.60	23.77	<0.001
Mental state	31.6	5.31	24.89	6.31	−6.75	−8.36	<0.001
Body assessment							
Sexual attractiveness	37.85	9.71	51.07	7.11	−13.22	−11.84	<0.001
Weight control	24.56	8.96	38.74	6.78	−14.18	−13.44	<0.001
Physical condition	24.82	7.40	36.63	4.98	−11.81	−14.37	<0.001

**Table 2 nutrients-16-00811-t002:** The significance of the differences between the means in the scope of the analyzed variables between the studied groups of women before the weight loss therapies.

Analyzed Variables	Group 1 N = 62		Group 2 N = 60		t/z *	*p*
	M	SD	M	SD		
Initial BMI	30.41	4.02	29.49	3.37	1.33 *	0.184
Mental state	25.45	7.01	25.48	5.77	0.17 *	0.866
Body assessment						
Sexual attractiveness	37.67	10.45	38.03	9.02	0.10 *	0.922
Weight control	23.90	9.56	25.19	8.36	−0.94 *	0.344
Physical condition	23.80	7.86	25.81	6.85	−1.49 *	0.135

* Mann–Whitney U test analysis result.

**Table 3 nutrients-16-00811-t003:** The significance of the differences between the means in the scope of the analyzed variables between the studied groups of women after the weight loss therapies.

Analyzed Variables	Group 1 N = 62		Group 2 N = 60		t/z *	*p*
	M	SD	M	SD		
Final BMI	22.54	1.88	22.16	1.91	1.08 *	0.282
Self-assessment	24.52	4.72	22.39	3.80	2.38 *	0.017
Mental state	10.18	8.45	16.53	9.46	−3.77 *	<0.001
Self-efficacy	32.08	4.87	31.21	5.71	0.62	0.533
Body assessment						
Sexual attractiveness	54.60	5.87	47.66	6.55	6.15	<0.001
Weight control	42.02	5.82	35.56	6.13	5.80 *	<0.001
Physical condition	38.87	3.97	34.47	4.92	5.10 *	<0.001

* Mann–Whitney U test analysis result.

**Table 4 nutrients-16-00811-t004:** Pearson’s r correlations between body assessment and the level of mental health (N = 122).

Analyzed Variables	Body Assessment		
	Sexual Attractiveness	Weight Control	Physical Condition
Level of mental health (first measurement)	−0.11	−0.02	−0.07
Level of mental health (second measurement)	−0.37	−0.42	−0.47

*p* < 0.005.

**Table 5 nutrients-16-00811-t005:** Results of stepwise regression analyses. General mental health was introduced as dependent variable; body assessment categories were introduced as independent variables (second measurement) (N = 122).

Dependent Variables	Independent Variables Which Reached Statistical Significance in the Model	Corr. R^2^	F	Beta	T	*p*
Level of mental health (second measurement)	Weight control, Physical condition	0.24	20.49	−0.23	−2.40	0.018
				−0.34	−3.63	<0.001

## Data Availability

The raw data supporting the conclusions of this article will be made available by the principal investigator upon reasonable request once all relevant substudies are reported and completed.

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
