# Peer review of "Mental Health and Body Image and the Reduction of Excess Body Weight in Woman (Polish Sample)"

_nutrients, 2024, doi:10.3390/nu16060811_

Round 1

Reviewer 1 Report

Comments and Suggestions for Authors

This is a well-designed clinical investigation,which emphsize the importance of diet and physical activity in the process of treating excess body weight .From the study, The culturally conditioned linking of low body weight with proper functioning is particularly evident nowadays. But I still have some questions for this assay.

1.the superviser have been readed in the protocol, but no more detail about the supervise can do.

2. In the study, should we collect more information ,besides BMI, to evaluate the excess weight loss.

3. No figure in this study, it may add a figure which is easily reflect the results.

Author Response

nasze odpowiedzi i wyjaśnienia do uwag Recenzenta 1

Reviewer 2 Report

Comments and Suggestions for Authors

The paper reports a longitudinal evaluation of psychological features in high-weight women who underwent a weight reduction protocol. The paper presents data that might be interesting for the readers but presents serious flows that must be considered in evaluating the manuscript:

- the statistical plan is not appropriate from my perspective. The aim of the manuscript required an evaluation of data as paired and a comparison of groups looking for the interactions between factors (timeXgroup). 

- The paper considered the pandemic effects but all the data were collected before 2020, so it is unclear to me the connections. 

- The time of the slim procedure and the adherence to the treatment are not reported or discussed. It is not clear to me the procedure protocol. How they screened 1000 people obtaining 122 participants for example. How people were followed during the period. How they evaluated if people performed different approaches than the one reported as the description of the group. 

- What is "mental health"? This is a very foggy term.

- The selection methods (participants recommended by other participants) could be a serious bias element

- The authors used sometimes the dot and sometimes the comma. Please, be consistent.

Author Response

for Reviewer 2

Round 2

Reviewer 2 Report

Comments and Suggestions for Authors

I'd thank the authors for their kind reply and for seriously taking my methodological concerns. I think their reply is adequate. I think the new version of the manuscript reports in a more balanced way their study. Looking at the manuscript, I still have a few comments for them: 

- were people screened for eating disorders? Have you asked if they performed any psychotherapy during the diet period?

- your sample used diet therapies to reduce their weight, is it possible that this procedure and the time needed could be protective against the difficulties in the update of body image representation? Because previous literature reported difficulties after bariatric surgery to update people's body image (see https://doi.org/10.1007/s11695-020-05166-z).

- please include in Figure 2 a graphical sign to underline the significant differences found.

Author Response

for the reviewer
